# From Scribbles to Script: Graphomotor Skills’ Impact on Spelling in Early Primary School

**DOI:** 10.3390/children10121886

**Published:** 2023-12-01

**Authors:** Michelle N. Maurer, Lidia Truxius, Judith Sägesser Wyss, Michael Eckhart

**Affiliations:** 1Institute for Special Needs Education, Bern University of Teacher Education, 3012 Bern, Switzerland; judith.saegesser@phbern.ch (J.S.W.); michael.eckhart@phbern.ch (M.E.); 2Department of Special Needs Education, University of Oslo, 0371 Oslo, Norway; 3Institute for Research, Development, and Evaluation, Bern University of Teacher Education, 3012 Bern, Switzerland; lidia.truxius@phbern.ch

**Keywords:** graphomotor skills, spelling, handwriting, working memory, motivation, gender, children

## Abstract

The acquisition of handwriting skills is a crucial goal in early primary school. Yet our comprehension of handwriting development, encompassing graphomotor skills and spelling, remains fragmented. The identification of predictors for handwriting skills is essential for providing early support. This longitudinal study aimed to explore the predictive roles of gender, working memory, and motivation to handwrite for graphomotor skills six months later and spelling skills one year later. Paper-and-pencil tasks (graphomotor skills, spelling), a tablet task (working memory), and a questionnaire (teachers’ ratings of children’s handwriting motivation) were employed. This study included 363 first-grade children (49.8% girls) aged 6–9 years. Results from a structural equation model, controlling for age and socioeconomic background, revealed that girls exhibited superior performance in graphomotor skills, while boys tended to spell more accurately. Furthermore, working memory predicted graphomotor skills but not spelling. Additionally, motivation to handwrite predicted both first-grade graphomotor skills and second-grade spelling. This study extends contemporary evidence, demonstrating that graphomotor skills predict spelling while considering gender and motivation. The findings underscore the pivotal role of graphomotor skills in spelling acquisition and suggest their contribution to spelling difficulties.

## 1. Introduction

### 1.1. A Developmental Perspective on Handwriting

Once children gain the ability to hold a pencil, they naturally attempt to represent the real world through visual forms. Graphomotor skills encompass a specific set of psychomotor abilities that enable drawing and handwriting. These skills involve coordinating cognitive processes with hand and finger movements to produce graphical representations. While drawing involves the act of creating or replicating images, handwriting focuses on forming letters, symbols, or figures. Writing, in comparison, is the translation of thoughts into linguistic representations at the word-, sentence-, and text-level. It is worth noting that, despite their conceptual differences, the terms *handwriting* and *writing* are frequently used interchangeably in the literature, leading to confusion when interpreting and distinguishing research outcomes related to these two concepts [1].

Developmental research on writing commonly distinguishes between two fundamental aspects of writing: transcription and text generation [2,3]. Transcription skills, often also referred to as handwriting, encompass the cognitive and physical act of creating written representations of text, involving graphomotor and spelling processes [4]. Handwriting involves the complex integration of cognitive and motor processes that early writers muster master. In contrast, text generation (i.e., writing) shares many language generation processes with oral language, such as content selection, lexical retrieval, and syntactic formulation, and is more a matter of generalizing from speech production to written production [4]. It is essential to emphasize that this study specifically focuses on handwriting skills, distinct from text generation (i.e., writing).

Handwriting builds the foundation for writing [5]. When children engage in drawing and handwriting activities, graphomotor skills play a crucial role. Graphomotor skills represent the prerequisites to handwrite, including visuomotor integration and fine motor control of finger movements to guide the pen precisely on the paper [6,7]. In addition to graphomotor processes, spelling retrieval processes are engaged once children write words [8]. Specifically, to write a letter, a child must retrieve the individual letter representation (i.e., grapheme) from memory, followed by retrieving the related shape of the letter (i.e., allograph). The allograph then drives the selection of a motor plan for generating the pen movements that lead to the letter on the page [7]. While graphomotor skills were found to predict broad measures of emergent literacy [9,10], only very few studies looked at their role in subsequent spelling [11]. Evidence from cross-sectional studies suggests that difficulties in graphomotor skills typically go along with difficulties in spelling skills [12,13] indicating an intricate relationship between graphomotor skills and spelling in early primary school.

Theoretical models of writing development further suggest that graphomotor skills and spelling interact with working memory [3]. Working memory plays a role in storing, maintaining, and manipulating verbal and visuo-spatial information. Theoretical assumptions in more proficient writers propose that among the various writing processes, planning requires visual-spatial working memory, while composing, reviewing, and revising text (including reading) requires verbal working memory [14]. During the early stages of learning to write, when children cannot yet generate text, visual-spatial working memory might be especially relevant for remembering and maintaining the geometric shape or letter shape depicted on the template and planning the execution (i.e., copy) of this shape. The interaction between graphomotor skills and working memory enables text generation in more proficient writers [3]. When graphomotor processes are not yet fluent, they require a sizable portion of the writer’s limited working memory capacity [4,15]. Once graphomotor skills become more proficient and automatic, working memory capacities are set free [16], which consequently results in better writing performance [4,15]. Working memory in turn can support spelling development [17,18], as spelling processes require considerable attentional resources as long as children engage in on-line construction of spellings (rather than automated retrieval of spellings) [4]. It is therefore likely to be assumed that graphomotor difficulties constrain working memory resources for spelling acquisition.

### 1.2. Individual Differences in Handwriting

Learning to handwrite is a complex task that is intensively practiced in the early school years. Unfortunately, children who struggle with handwriting acquisition often go unnoticed and lack sufficient attention and support [6,19]. One possible explanation could lie in the tendency of current diagnostic and educational systems to prioritize writing over the handwriting process, failing to adequately address the graphomotor difficulties that may hinder the acquisition of efficient, legible, and grammatically correct handwriting. This highlights a significant gap, emphasizing the need to recognize and tackle the specific hurdles associated with handwriting acquisition. It is surprising that this lack of attention persists, given that graphomotor difficulties can significantly restrict writing and reading and therefore hamper learning at school fundamentally [10,11,20,21]. To illustrate, consider a child who faces challenges with graphomotor skills while writing a paragraph on ravens. Without appropriate adaptations of the task for this particular child, the child is likely to encounter greater difficulties in producing correctly spelled content and may also acquire less knowledge about paragraph structure and the subject of ravens compared to a child with proficient graphomotor skills. Similarly, a child who struggles to produce legible and neatly aligned digits will most likely need to invest more time and effort in math homework while learning less, compared to a child with more proficient graphomotor skills [19]. This illustrates the importance of handwriting didactics that are adequate for students’ heterogeneous needs [22]. 

Certainly, children struggling with handwriting are not isolated cases. The prevalence of handwriting difficulties varies, ranging from 7% to 34%, depending on factors such as criteria used, assessment method, country, the children’s age, and the types of raters involved, which can include psychologists, therapists, teachers, and researchers [23,24,25]. Notably, among the children facing handwriting difficulties, boys tend to be disproportionately affected. Research on primary and secondary school-aged children typically indicates that boys exhibit less proficient handwriting compared to girls [26,27,28]. However, contrasting findings revealing no significant variations in handwriting skills between boys and girls underscore the necessity for further investigations [29,30]. The underlying reasons for gender differences in handwriting have received limited exploration, but see [31]. Nonetheless, considering evidence indicating more proficient fine motor skills in girls compared to boys [32], it can be hypothesized that individual differences in the extent to which fine motor and visuomotor integration skills are practiced may play a role. It has been argued that girls, in comparison to boys, often receive greater encouragement from their surroundings to engage in fine motor activities [33,34]. This assumption might partly explain why girls typically gain more experience in fine motor and visuomotor integration skills and tend to outperform boys in handwriting, as observed in previous studies, e.g., [27,31,32,35].

An additional factor likely contributing to individual differences in the extent and duration of children’s engagement in paper and pencil activities is their motivation. Mekyska and colleagues [1] recently speculated about individual differences in children’s motivation to produce neat and tidy handwriting. Furthermore, Feder and Majnemer [36] mentioned motivational factors in handwriting. However, the impact of a child’s motivation for handwriting on their graphomotor and spelling development has not yet been investigated and consequently remains poorly understood. Studies examining more proficient writers argue that motivational beliefs influence the extent to which one engages in writing, the level of effort committed, and how students interact with others, such as teachers or peers [37]. These motivational beliefs have been shown to influence writers and their written products [38,39]. It is reasonable to assume that, not only in more proficient writing but also in early handwriting development, individual differences in handwriting motivation influence how often children draw and handwrite, consequently impacting their handwriting development and proficiency. Given the scarcity of research on early handwriting motivation, we included handwriting motivation as an exploratory measure in this study.

### 1.3. The Present Study

This longitudinal study of early primary school children builds upon contemporary writing models and evidence, aiming to contribute to the limited understanding of handwriting development in young handwriters. By disentangling graphomotor and spelling skills and identifying their respective predictors, namely gender, working memory, and motivation for handwriting, this study seeks a more nuanced understanding of why some children encounter greater challenges in handwriting than others. The research questions (RQ) guiding this study are as follows:RQ1: Do gender, visual-spatial working memory, and motivation to handwrite at the beginning of first grade predict graphomotor skills at the end of first grade?RQ2: Do gender, visual-spatial working memory, and motivation to handwrite at the beginning of first grade predict spelling skills at the beginning of second grade?RQ3: Do graphomotor skills at the end of first grade predict spelling skills at the beginning of second grade?

## 2. Materials and Methods

### 2.1. Participants

A total of *n* = 363 children from urban (*n* = 239) and rural (*n* = 124) areas of the German-speaking region of Switzerland were tested in three waves with an interval of six months between each wave: at the beginning of first grade (T1; October to December), end of first grade (T2; April to June), and beginning of second grade (T3; October to December). At the first measurement (T1), the children had a mean age of seven years (*SD* = 4.65 months, range: 6 years 3 months to 8 years 2 months). The gender distribution in the sample was well-balanced, with 49.8% girls. However, the boys in the sample were slightly older than the girls (F(1361) = 11.14, *p* < 0.001, η = 0.03). This minor age difference may be attributed to the perception that boys are often considered less prepared for school by parents and teachers, leading to a tendency for them to enter school later [40]. By the start of first grade, all children had established a hand preference for writing, with 86% of the children being right-handed. 

Approximately 40% of the children in this study received one or more forms of additional educational support. In Switzerland, additional educational support is often low-threshold and assists children who may not yet have reached distinct learning goals. Within our sample, this additional educational support included special needs support (*n* = 73), speech and language therapy (*n* = 40), psychomotor therapy (*n* = 37), special language support for non-native German speakers (*n* = 37), and occupational therapy (*n* = 8). Most children’s first language was Swiss German/German (73.83%). To estimate the children’s socioeconomic background, their parents’ occupation was assessed using the International Socio-Economic Index of Occupational Status (ISEI; [41]. The ISEI considers occupational prestige with respect to income and education. The range of socioeconomic backgrounds was diverse, with scores spanning from 14.39 (indicating occupations such as waste disposal workers) to 88.96 (representing high-status professions like judges). The mean occupational prestige for the children’s mothers was *M* = 54.60 (*SD* = 20.05), and for the fathers, it was *M* = 56.88 (*SD* = 21.63), which is slightly higher than the European mean [42].

Parents provided written content for their children’s participation before the first measurement, and the children themselves provided verbal consent for participation. This study was part of a bigger research project and was approved by the Ethics Committee of the Faculty of Human Sciences of the University of Bern, Switzerland (Approval No. 2020-10-00005).

### 2.2. Materials

#### 2.2.1. Graphomotor Skills

We used the graphomotor screening GRAFOS-2 [6,43] to assess children’s capacity to replicate various geometric shapes that are fundamental elements of letter writing (e.g., circles, squares, triangles, crosses) and more complex geometric shapes (e.g., rhombus, connected loops, lying eight). Accurate copying of these shapes demands visuomotor integration and fine motor skills, which are the central elements of graphomotor skills. The GRAFOS-2 screening is embedded within a cover story, wherein children are instructed to copy 12 different shapes, each shape six times as precisely as possible, in predefined fields of 1 cm^2^ on the screening sheet. To evaluate the accuracy, each shape (72 copies in total) was assessed against specific criteria, with a rating of 1 denoting an accurate reproduction and 0 indicating an inaccurate reproduction. For example, the criteria for an accurate reproduction of a circle included the circle being closed and without any “corners”. The internal consistency assessed using Cronbach’s Alpha was high (α = 0.80 for the eight fundamental shapes, α = 0.78 for the four complex shapes). 

#### 2.2.2. Spelling

To measure children’s spelling skills, children were asked to spell four isolated words selected from the Hamburg Writing Test (German: Hamburger Schreib-Probe) [44]. These words were chosen in collaboration with teachers and linguists, so they are suitable for children in the first and second grades. The selected four words varied in orthographic complexity, containing graphemes that require not only letter knowledge but also orthographic understanding. The Hamburg Writing Test is a well-established instrument that is frequently used to assess early literacy skills in beginning writers (e.g., [45,46]). Each word was verbally presented and accompanied by a corresponding picture to enhance comprehension and visualization. Given the varying lengths of the words, spelling proficiency was quantified by calculating the percentage of correctly spelled graphemes for each word.

#### 2.2.3. Motivation

At the beginning of the first grade, teachers were asked to assess children’s motivation for drawing and handwriting using a three-point Likert scale, where 1 denoted low motivation, 2 represented average motivation, and 3 indicated high motivation. Teachers rated the motivation levels of 45.8% of the children as high (girls: 54.6%; boys: 37.5%), 44.7% as average (girls: 41.4%; boys: 47.8%), and 9.5% as low (girls: 4.0%; boys: 14.6%). A Mann–Whitney U test confirmed that boys’ motivation was rated significantly lower than girls’ motivation (*U* = 12,607, *p* < 0.001, *r* = 0.21).

#### 2.2.4. Visual-Spatial Working Memory

We assessed children’s visual-spatial working memory using the Position Span Task [47], which is a child-adapted version of the Corsi-Block Tapping Task [48], a well-established instrument to assess young children’s working memory (e.g., [49,50]). This task was presented on a laptop computer equipped with a touch screen and audio instructions. In this task, a mole appeared at various locations within a 4 × 4 grid, following a predetermined pseudo-randomized pattern. The children’s task was to remember the locations where the mole had appeared and then touch those specific fields in reverse order after a delay of 1000 milliseconds. Each mole appearance lasted for 1200 milliseconds, with a 500-millisecond pause when the empty grid remained visible. The task began with a sequence of two mole appearances and increased in complexity by adding one mole appearance when the child correctly recalled at least three out of six sequences for each span length. The task terminated when the child incorrectly recalled more than three sequences within a particular span length. For the analysis, we used the total number of correctly remembered sequences across all span lengths.

### 2.3. Statistical Analyses

For the investigation of longitudinal predictors of graphomotor and spelling skills, we performed a structural equation model with a maximum likelihood estimation using Mplus [51]. At the beginning of first grade (T1), we included gender, working memory, and motivation to handwrite as manifest predictor variables to assess their direct and indirect effects through graphomotor skills on spelling at the beginning of second grade (T3). Graphomotor skills at the end of first grade served both as an outcome variable of the T1 predictors and as a predictor variable for spelling (T3). The predictor variables at T1 were correlated. Additionally, we controlled for children’s age and socioeconomic background (modeled as latent variables incorporating mother’s and father’s ISEI scores) while estimating paths to all predictor and outcome variables. Model fit was evaluated according to the criteria of Hu and Bentler [52], with CFI > 0.95, RMSEA < 0.06, and SRMR < 0.08 indicating a good model fit. 

## 3. Results

### Predictors of Handwriting Development

To receive an overview of the relationship among all included variables, we first calculated correlations between the control, predictor, and outcome variables (see Table 1). Motivation was positively correlated with both fundamental and complex graphomotor skills, as well as spelling and working memory. Graphomotor skills were associated with spelling but not with working memory, and spelling was associated with working memory. 

In a subsequent step, we calculated a structural equation model to investigate the longitudinal effects of gender, working memory, and motivation to handwrite at the beginning of first grade on graphomotor and spelling skills at the end of first grade and beginning of second grade, respectively, while controlling for children’s age and socioeconomic background. Figure 1 shows the significant paths of the structural equation model. The model showed an excellent fit: CFI = 1.00, RMSEA = 0.01, SRMR = 0.03. The results revealed that children’s motivation to handwrite significantly predicted first-grade graphomotor skills and second-grade spelling. Gender was found to predict graphomotor skills and spelling, although in different directions, with girls displaying an advantage in graphomotor skills and boys in spelling. Working memory uniquely predicted graphomotor skills but did not predict spelling skills. Furthermore, girls tended to show higher motivation to handwrite compared to boys, while a higher motivation to handwrite seemed to go along with higher working memory capacity. In total, the predictors explained 19.5% of the variance in first-grade graphomotor skills and 21.5% of the variance in second-grade spelling. Crucially, graphomotor skills meaningfully predicted later spelling (β = 0.31).

## 4. Discussion

This longitudinal study aimed to investigate the predictive contributions of gender, visual-spatial working memory, and motivation to handwrite on subsequent graphomotor skills and spelling. Additionally, this study investigated the predictive role of first-grade graphomotor skills on second-grade spelling skills. The overarching objective was to clarify how individual differences in these predictors contribute to the challenges some children face in handwriting.

### 4.1. The Role of Gender, Working Memory, and Motivation for Subsequent Handwriting

The results concerning gender as a predictor revealed a significant advantage for girls over boys in developing more proficient graphomotor skills, even after controlling for age and socioeconomic background. In contrast to graphomotor skills, being a boy as opposed to a girl was advantageous for developing spelling skills. The predictive effect of gender persisted, even when gender was correlated with motivation (girls showed higher motivation than boys). These findings are somewhat consistent with recent research reporting distinct correlates of handwriting for girls and boys [31]. It is possible that different pathways to acquire spelling skills exist and that boys in the sample might compensate for their lower graphomotor skills. Nonetheless, the processes underlying these potentially different pathways to spelling warrant further investigation. 

Consistent with contemporary models of writing [3,14], it was observed that visual-spatial working memory predicted graphomotor skills six months later. This association can be attributed to the necessity of working memory capacity in the process of maintaining the letter shape and initiating a motor plan for writing the specific letter on paper, particularly in non-automated handwriting [7,53]. The more automated and fluent the graphomotor skills, the lower the cognitive load. In other words, higher working memory capacity might facilitate the complex acquisition of graphomotor processes. However, in contrast to theoretical expectations that spelling acquisition requires substantial attentional resources [4], working memory did not predict spelling skills one year later. Given the links between working memory and subsequent graphomotor skills and between graphomotor skills and subsequent spelling, it is reasonable to assume that graphomotor skills mediate the longitudinal effect of working memory on spelling. 

The results concerning motivation as a predictor revealed that children who displayed higher motivation for handwriting at the beginning of first grade, as reported by their teachers, demonstrated greater proficiency in graphomotor skills by the end of first grade and more advanced spelling skills at the beginning of second grade. These effects were of moderate magnitude [54]. These findings underscore the significance of teachers and parents nurturing children’s motivation and enthusiasm for handwriting activities [55]. This is particularly crucial for children with low motivation, a group that predominantly consisted of boys in our sample. Low motivation is likely linked to reduced practice, creating a cycle that exacerbates performance disparities and potentially contributes to increased avoidance of handwriting among struggling children. 

An intriguing question to consider is the extent to which teachers’ beliefs and expectations regarding their students’ handwriting motivation impact students’ handwriting outcomes. As the functioning and disability of an individual, including handwriting, always occur in context, it is important to take environmental factors affecting child development into account. In comparison to the DSM-5 (Diagnostic and Statistical Manual of Mental Disorders, Fifth Edition) and the ICD-11 (International Classification of Diseases, 11th Revision) classification systems, the ICF-CY (International Classification of Functioning, Disability, and Health for Children and Youth) of the WHO (World Health Organization) does not focus on specific diagnoses and criteria but instead places emphasis on environments that allow all children, irrespective of their abilities, to participate in education. Following this line of thought, it could be argued that teachers may interact differently with students they perceive as “unmotivated,” which could directly or indirectly influence children’s handwriting development. The substantial impact of teachers’ expectations on student achievement has been previously demonstrated, as exemplified by the research of Rosenthal and Jacobson in 1968 [56] and subsequent studies on the so-called Pygmalion effect [57]. In addition to teachers’ expectations, their attitudes towards diverse classrooms and the instructional methods used also significantly affect students’ handwriting development [22]. When learning materials are suitable for the varying abilities within a classroom, it becomes more likely to maintain students’ handwriting motivation [58].

### 4.2. Graphomotor Skills as a Foundation for Spelling

Well-established theoretical frameworks of writing development [3] and previous empirical evidence underscore the interconnected nature of graphomotor skills and spelling, with both cross-sectional associations (e.g., [12,18]) and longitudinal associations (e.g., [11,59]) supporting this notion. Our study advances previous findings by specifically demonstrating that graphomotor skills serve as a predictor for later spelling proficiency in this sample of emerging writers. The strong impact of first-grade graphomotor skills on second-grade spelling, as observed in this sample, implies that difficulties in graphomotor skills may limit the cognitive resources accessible for acquiring accurate spelling abilities. Essentially, as graphomotor skills become more proficient and automatic, working memory resources become liberated and can be directed towards spelling. This is crucial because, once spelling retrieval becomes automatic, resource demands associated with spelling are minimized, thereby freeing up cognitive resources for text generation in more proficient writing (e.g., [60]).

In this sample of beginning writers, the children facing the greatest graphomotor challenges were predominantly boys, individuals with lower working memory capacity, and those with low motivation to handwrite, also predominantly boys. Additionally, graphomotor skills were found to predict spelling skills six months later. These findings underscore the pivotal role of graphomotor skills as a foundational element for spelling proficiency, enhancing our understanding of why certain children encounter more difficulties in developing proficient handwriting skills than others. The results of this study hold practical implications, encouraging pedagogical approaches that effectively address graphomotor skills as a crucial foundation for spelling, which collectively facilitate higher-order writing. By offering learning activities suitable for diverse and heterogeneous classrooms, most children can develop a solid basis of graphomotor skills, and importantly, children who face challenges are more likely to stay motivated. 

### 4.3. Limitations

While this study provides valuable insights into handwriting development in first-grade children, it is crucial to acknowledge several limitations when interpreting the results. Notably, the measure of children’s handwriting motivation is exploratory, and the reliability and validity of the teacher’s rating on a 3-point Likert scale are limited. Despite these constraints, given the scarcity of research on handwriting motivation in young children, this study adopted an initial exploratory approach. Future studies are encouraged to directly measure handwriting motivation in children rather than relying on teacher assessments and to comprehensively assess handwriting motivation. 

Additionally, while motivation to handwrite was explored as a predictor of subsequent graphomotor and spelling skills, it is plausible that reciprocal associations between motivation and graphomotor skills, as well as spelling, more accurately depict their relationships. In this study, primarily focused on predictors of handwriting development, motivation to handwrite was exclusively assessed at the initial measurement point. To gain a more comprehensive understanding of the developmental trajectories and causal relationships between motivation and handwriting, future studies should explore the development of motivation to handwrite, graphomotor skills, and spelling across multiple time points. Such an approach will assist in disentangling the intricate relationships involved. Finally, it is important to acknowledge that additional unexamined variables may explain individual differences in handwriting development. Depending on the (number of) predictor variables considered in the model, the variance explained by each predictor may vary, potentially yielding different results.

## 5. Conclusions

Children who face greater challenges with graphomotor skills in the first grade tend to exhibit less accurate spelling skills in the second grade. Graphomotor skills and spelling, the two integral components of handwriting, share some predictors (e.g., gender, motivation), but also have unique predictors (e.g., working memory) influencing them. Notably, both motivation and gender emerged as predictors for subsequent graphomotor and spelling skills, while working memory was specifically associated with subsequent graphomotor skills. Furthermore, it is worth mentioning that girls showed an advantage over boys in developing graphomotor skills, while boys had an advantage over girls in developing more proficient spelling skills. Even after accounting for gender and motivation, graphomotor skills meaningfully predicted spelling, highlighting the importance of graphomotor skills in the acquisition of spelling proficiency.

## Figures and Tables

**Figure 1 children-10-01886-f001:**
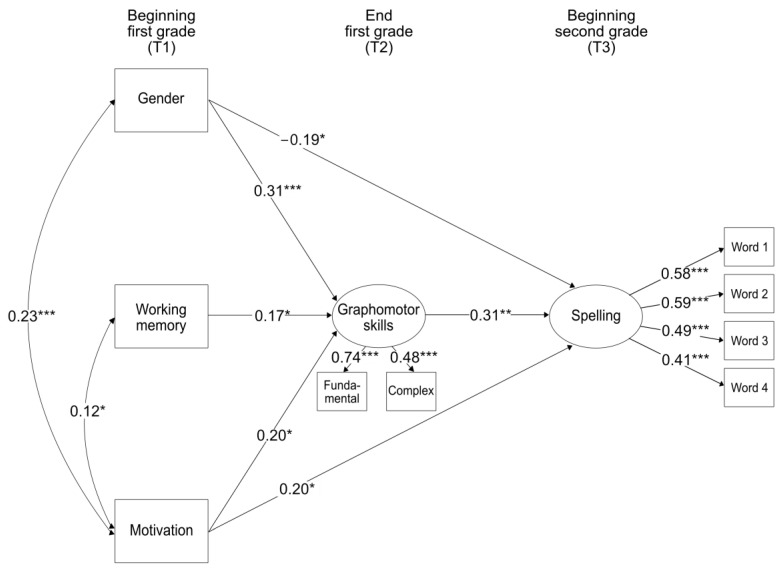
Structural equation model predicting graphomotor and spelling skills. Bidirectional arrows show concurrent correlations and unidirectional arrows show longitudinal effects of the constructs and factor loadings of graphomotor and spelling skills. The results displayed are controlled for children’s age and socioeconomic background. Standardized coefficients are shown. Gender: 1 = boys; 2 = girls; * *p* < 0.05; ** *p* < 0.01; *** *p* < 0.001.

**Table 1 children-10-01886-t001:** Pearson correlations among all variables.

	1	2	3	4	5	6	7
1. Age	-						
2. Socioeconomic background of the mother	−0.04	-					
3. Socioeconomic background of the father	0.00	0.43 ***	-				
4. Motivation ^a^	−0.03	0.17 **	0.19 ***	-			
5. Working memory	−0.04	0.00	0.10	0.15 **	-		
6. Graphomotor skills (fundamental)	−0.01	−0.03	−0.03	0.11 **	0.00	-	
7. Graphomotor skills (complex)	−0.03	−0.03	0.05	0.24 ***	0.05	0.36 ***	-
8. Spelling	−0.05	0.08	0.08	0.22 ***	0.18 ***	0.15 **	0.12 *

Notes. ^a^ Spearman correlations: * *p* < 0.05; ** *p* < 0.01; *** *p* < 0.001.

## Data Availability

The data presented in this study are openly available in the Open Access Repository at Bern University of Teacher Education (REPO PHBern) at https://doi.org/10.57694/7097 (accessed on 28 October 2023).

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
