# Peer review of "From Scribbles to Script: Graphomotor Skills’ Impact on Spelling in Early Primary School"

_children, 2023, doi:10.3390/children10121886_

Round 1
Reviewer 1 Report
Comments and Suggestions for Authors
The manuscript presents interesting work on the relationship between graphomotor skills and spelling with other variables such as gender and other visuospatial and motivational variables.
The authors are congratulated for the study.
Some improvements are recommended:
TITLE AND SUMMARY
1. The title should be concise and not a summary of the work
2. The summary must be improved. The information that must appear are: brief introduction (two sentences), objective of the study, methodology (sample, variables and instruments, and procedure), results and discussion (educational implications)
THEORIC INTRODUCTION
3. The authors do not adequately justify why they focus on investigating visuospatial perception, motivation and gender in graphomotor skills and spelling, over other variables that are also related. Why do you select these three variables over others?
4. The title does not indicate that the study addresses difficulties in spelling or psychomotor development, but an entire section is dedicated to it and the sample also includes students with special educational needs. This must be corrected.
5. The section on writing difficulties includes dysorthography, but not dysgraphia, an interesting issue when we talk about psychomotor problems or handwriting.
6. The other two variables of gender and motivation are indicated anecdotally and briefly at the end of the theoretical introduction, without entering into a relationship or depth with the other variables of writing and psychomotor skills. This section needs to be strengthened with a greater number of investigations.
7. The authors add another section “the present study” where the general objective is not included. Three research questions are raised, which are not clearly reflected in the results or in the discussion.
METHOD
participants
8. The authors should provide more data on the sample, describe the sociodemographic and cultural characteristics of the participants.
9. Almost half of the sample has some special educational need, which cannot be due to chance, this must be clear in the objectives of the study.
10. Part of the results included should be included in the description of the sample (Age, sociocultural, etc.). They must explain why there are differences in age between genders, since the differences are significant.
materials
11. The authors do not offer information on spelling tests, and why do they use four words? which are? What properties do those four words have? Why do they use that measure of proportion of success, and not a statistical average between the participants.
12. The motivation variable is also not well described or measured. What definition of motivation and what measure have they given? What reliability and validity does this measure have?
13. Validity and reliability information for the Position Span Task test must be indicated.
Data design, procediments and analysis
14. A design section should be included, where the variables are specified, whether it is longitudinal, research groups according to gender or level of motivation, etc.
15. Include statistical analyzes according to the research questions or objectives
16. The effect size test, which later appears in the results section, and its interpretation are not included.
17. Ethical standards and evaluation procedures must be included.
RESULTS
18. Section 3.1 includes sample description information
19. If the percentage of success is finally maintained as a measure of spelling, an F cannot be applied
20. The following section should include the aforementioned correlations, and not in an appendix
21. Finally the structural models
22. In Figure 1 it is not understood that the correlation between the spelling variable and the four words that make it up is indicated at the end, it is unnecessary and leads to confusion.
DISCUSSION
23. The discussion must be organized to the objective and research questions. It is not like this.
24. The authors point out that there are differences in spelling between genders, when there are no significant differences (line 260)
25. The explanation at the end of the first paragraph may not be clear (lines 268-271) when pointing out that psychomotor and spelling skills may not be related and there are different mechanisms for psychomotor skills and for spelling. That is, they are different concepts, on the one hand, difficulties in psychomotor skills and on the other hand, difficulties in spelling, and some children are comorbid. That is, they are different concepts.
26. The paragraph on motivation could be relevant (line 280 et seq.), although it is more of a limitation of the study. Motivation has been evaluated through the teacher's opinion and not with some more direct measure, so it may be that the teacher has not made reference to motivation, but rather to the student's ability, or his expectations about achievement. of the child, etc.
27. The discussion and conclusion section
CITES AND REFERENCES
28. The authors use APA version 7 and it does not conform to the journal's standards. They must review and be careful.
Reviewer 2 Report
Comments and Suggestions for Authors
Sample- inform about ethical procedures
lines 174-6- specify criteria for accurate and inaccurate
lines 184-6- specify criteria for spelling proficiency
lines 188-190- include references to sustain this instrument
Results
identify, estimate and present effect size for U
table 2- include percentages
Reviewer 3 Report
Comments and Suggestions for Authors
Dear authors,
Please find enclosed the review of your article.
Sincerely

Author Response
Please see the answers in the attachment. Thank you for your in-depth feedback.

Round 2
Reviewer 1 Report
Comments and Suggestions for Authors
Dear editor,
the manuscript has been substantially improved and is suitable for publication
Best Regards,
Reviewer 3 Report
Comments and Suggestions for Authors
Thanks